# Possible Association between Genetic Diversity of Hepatitis B Virus and Its Effect on the Detection Rate of Hepatitis B Virus DNA in the Placenta and Fetus

**DOI:** 10.3390/v15081729

**Published:** 2023-08-12

**Authors:** Sirinart Sirilert, Pattara Khamrin, Kattareeya Kumthip, Rungnapa Malasao, Niwat Maneekarn, Theera Tongsong

**Affiliations:** 1Department of Obstetrics and Gynecology, Faculty of Medicine, Chiang Mai University, Chiang Mai 50200, Thailand; sirinart.s@cmu.ac.th; 2Department of Microbiology, Faculty of Medicine, Chiang Mai University, Chiang Mai 50200, Thailand; pkhamrin@gmail.com (P.K.); kattareeya.k@cmu.ac.th (K.K.); nmaneeka@gmail.com (N.M.); 3Department of Community Medicine, Faculty of Medicine, Chiang Mai University, Chiang Mai 50200, Thailand; rungnapa.m@cmu.ac.th

**Keywords:** fetal HBV exposure, genotype, hepatitis B virus, intrauterine HBV exposure, placental HBV exposure, subgenotype

## Abstract

***Background***: The prevalence of HBV infection and HBV genotypes varies from country to country, and the role of HBV genotypes in the presence of HBV in the placenta and fetus has never been explored. This study was conducted to (1) identify HBV genotypes, and their frequencies, that infected Northern Thai pregnant women; (2) evaluate the association between HBV genotypes and the detection rate of HBV DNA in the placenta and fetus; (3) evaluate the association between specific mutations of the HBV genome and HBV DNA detection in placental tissue; and (4) identify the mutation of the HBV genome that might occur between maternal blood, placenta, and cord blood. ***Methods***: Stored samples of the maternal blood, placental tissue, and cord blood that were collected from 145 HBsAg-positive pregnant Thai women were analyzed to identify HBV DNA. ***Results***: Approximately 25% of infected mothers had fetal HBV DNA detection, including cases with concomitant HBV DNA detection in the placenta (77.3%). A total of 11.7% of cases with placental detection had no HBV DNA detection in the maternal blood, indicating that the placenta could be a site of HBV accumulation. Of the 31 HBV-positive blood samples detected by nested PCR, the detected strains were subgenotype C1 (77.4%), subgenotype B9 (9.7%), and subgenotype C2, B2, B4, and recombinant B4/C2 (3.2% for each). Genotype B had a trend in increased risk of placental HBV DNA detection compared to genotype C, with a relative risk of 1.40 (95% CI: 1.07–1.84). No specific point mutation had a significant effect on HBV DNA detection in placental tissue. Mutation of C454T tended to enhance HBV DNA detection in placental tissue, whereas T400A tended to have a lower detection rate. No mutation was detected in different sample types collected from the same cases. ***Conclusions***: HBV DNA detection in the fetus was identified in approximately 25% of HBV-positive mothers, associated with the presence of HBV in the placenta in most cases. The placenta could possibly be a site of HBV accumulation. Subgenotype C1 was the most common subgenotype, followed by subgenotype B9. HBV genotype B possibly had a higher trend in intrauterine detection than HBV genotype C. Mutation is unlikely to occur during intrauterine exposure.

## 1. Introduction

Hepatitis B virus (HBV), a bloodborne pathogen, is a DNA virus belonging to the genus Orthohepadnavirus of the Hepadnaviridae family. HBV has a lipid envelope and an icosahedral nucleocapsid core composed of a core protein. It has approximately 3.2 kb double-stranded circular DNA genome containing four overlapping open reading frames (ORFs), which encode the surface protein (S), core protein (C), polymerase (P), and x protein (X) [1,2]. After attachment to host cell membrane, the HBV genome can integrate into the host genome, then start the replication cycle [2]. The remarkable genetic diversity of the HBV genome is due to the lack of proofreading activity of its DNA polymerase [3,4]. To date, human HBV is classified into at least ten genotypes (A–J) based on a difference in the nucleotide sequence of the whole genome or of the S-region of more than 8% or 4%, respectively [4,5,6]. Also, HBV has an evolutionary process during chronic infection under immune pressure called mutation–selection–adaptation. HBV replicates through an RNA intermediate and can integrate into the host genome. The unique features of the HBV replication cycle confer a distinct ability of the virus to persist in infected cells. Virological and serological assays have been developed for the diagnosis of various forms of HBV-associated diseases and for the treatment of chronic hepatitis B infection.

The route of HBV vertical transmission from exposure to maternal blood at the time of delivery is considered the most common route of transmission in endemic areas [7,8,9,10]. Based on the association between the prevalence and route of transmission, it was reported that mother to child transmission or vertical transmission is the major transmission route in the areas of high prevalence (>7%), whereas sexual transmission is the major transmission route in low-prevalence areas (<2%) [11]. More importantly, up to 90% of infected newborns via vertical transmission from HBV-infected mothers during the perinatal period become chronic carriers [12,13], whereas only 5–10% of infected cases caused by transmission via other routes, such as sexual or blood contacts in adults, become chronic carriers [7]. To prevent vertical transmission, universal antenatal HBV screening has been suggested to identify HBV infection status at the first visit to a prenatal care unit, and it is recommended that HBV immunoglobulin together with HBV vaccine within 12 h after birth must be administered to all newborns of HBsAg-positive mothers as dual immunoprophylaxis. Before the era of antenatal maternal HBV screening and neonatal passive and active immunization, the vertical transmission rate of HBeAg-positive women was 70–90% and that of HBeAg-negative women was 25% [14,15].

The successful implementation of the guidelines significantly reduced the prevalence of HBsAg-positive children under the age of five years to 1.3% worldwide [16,17]. Nevertheless, 50 million new cases of HBV infection are still diagnosed every year. The majority of these new cases are caused by breakthrough infection and the failure of dual neonatal immunoprophylaxis, which result in mother-to-child transmission [18]. Furthermore, the current failure rate is higher than expected. Overwhelming evidence indicates that a high level of HBV viral load is a high risk factor for mother-to-child transmission and is strongly associated with immunoprophylaxis failure in children born to HBV-infected mothers [19]. Immunoprophylaxis failure can occur in about 10–30% of infants born to mothers with high HBV DNA levels of greater than 1,000,000 copies/mL or 200,000 IU/mL [20]. Therefore, antiviral drug administration in the third trimester to reduce the HBV viral load has been introduced as an additional intervention to prevent immunoprophylaxis failure and has been widely accepted [19]. Accordingly, antiviral drug administration during pregnancy has been recommended in pregnant women with a high HBV DNA level of greater than 1,000,000 copies/mL or 200,000 IU/mL [21,22,23,24,25,26,27,28]. Maternal antiviral drugs, together with neonatal immunoprophylaxis administration, may reduce HBV persistent infection in infants to less than 1% [18]. In Thailand, we have the guideline of routine screening of all pregnant women at the first visit and neonatal dual immunoprophylaxis, as well as selective antiviral drug administration in cases of HBV DNA > 2,000,000 IU/mL from 24 to 28 weeks of gestation to one month after birth, although the guideline might not be followed perfectly in practice.

Although the strategy of universal HBsAg screening, neonatal immunoprophylaxis for HBsAg-positive mothers, and selective use of maternal antiviral drugs is highly effective in the prevention of vertical transmission, the true effectiveness in a wide range of implementations needs to be investigated. Several other hidden factors must be taken into considerations, such as different HBV genotypes in different geographical areas, laboratory capability of viral load measurement, and the unavailability of antiviral drugs in low-resource settings, etc. A recent meta-analysis showed that HBeAg testing can be used as an alternative method for viral load measurements since it has high sensitivity (88%) and specificity (93%) in identifying cases with a high viral load greater than 200,000 IU/mL; it also has high sensitivity (99.5%) in predicting failure of neonatal immunoprophylaxis, even though the specificity (62.2%) is not very high [29]. Moreover, viral factors other than viral load, such as HBV genotypes, that might play a role in various degrees of placental and fetal detections, at least in part, need to be explored. In summary, several other hidden factors to empower the strategy to prevent mother-to-child transmission of HBV still need to be investigated to achieve the goals set by the WHO, including a 30% decrease in new cases or 1% prevalence of HBsAg-positive children in 2020 and a 90% reduction in new cases or 0.1% prevalence of HBsAg-positive children in 2030 [16]. 

Accordingly, the aims of this study were as follows: (1) to identify HBV genotypes, as well as their frequencies, that infect Northern Thai pregnant women; (2) to evaluate the association between HBV genotypes and the detection rate of HBV DNA in the placenta and fetus; (3) to evaluate the association between specific mutations of the HBV genome and placental HBV DNA detection; and (4) to identify the mutation of the HBV genome that may occur between maternal blood, the placenta, and cord blood.

## 2. Patients and Methods

A cross-sectional study was conducted on stored maternal blood, placental tissue, and cord blood samples collected from HBsAg-positive pregnant women giving birth at Maharaj Nakorn Chiang Mai Hospital between August 2017 and September 2020, with ethical approval by the Institutional Review Boards (Faculty of Medicine, Chiang Mai University; Study code: OBG-2564-08142). The samples were collected from the enrolled women with the following inclusion criteria: (1) singleton pregnancy; (2) a positive test for HBsAg during routine screening at the first antenatal care visit; and (3) delivery at Maharaj Nakorn Chiang Mai Hospital. Pregnant women with other underlying diseases such as HIV co-infection and cases with incomplete clinical data were excluded. Samples from 145 pregnant women with HBV infection were recruited in this study.

### 2.1. Sample Collection Methods

Sampling of the maternal venous blood, fetal (umbilical cord) blood, and placental tissue was performed in all cases. Maternal blood samples were taken at intrapartum after admission to the labor room, while fetal umbilical cord blood and placental tissue samples were collected shortly after birth. Maternal blood samples were collected in EDTA tubes at the time of intravenous fluid accession. After the fetus was delivered and sent to the pediatricians, fetal blood was taken from the umbilical cord under aseptic conditions, using normal saline and povidone-iodine to absolutely avoid maternal blood contamination. Both maternal and cord blood samples were collected in EDTA tubes and stored at 4 °C for subsequent HBV detection and molecular characterization. Placental tissue at the area under the umbilical cord insertion was cut into small pieces of approximately one cubic centimeter and kept in clean plastic bags. For HBV genome extraction, 25 mg of each placental tissue sample was washed with phosphate-buffered saline (PBS) to completely remove maternal blood contamination. Then, DNA extraction was conducted immediately after washing. Maternal and cord blood samples were centrifuged, and the plasma was collected and kept in aliquots of 1 mL. Placental tissue samples were collected in plastic bag containers. All the samples were stored in the freezer at a temperature of at least −20 °C.

### 2.2. HBV DNA Processing Methods

*(1) Viral genomic DNA extraction:* Viral DNA genome was extracted from 200 µL of maternal and cord plasma using a Geneaid™ DNA Isolation Kit (Geneaid Biotech Ltd., New Taipei City, Taiwan) according to the manufacturer’s instruction. Additionally, viral genomic DNA from the placental tissue was also extracted from 25 mg of tissue after cleaning with phosphate-buffered saline (PBS) until no maternal blood contamination was observed. The extraction was conducted using NucleoSpin^®^ Tissue from MACHEREY-NAGEL GmbH and Co. KG, Dueren, Germany following the manufacturer’s protocol. All extracted DNA was stored at −80 °C until further use.

*(2) DNA amplification:* Conventional PCR was conducted to amplify the genome of HBV using the primers designed and tested with the standard primer sets reported by Sagnelli et al. [30]. First, HBV DNA was detected by nested PCR. The S gene was amplified using the protocol and primer sets as described by Sagnelli et al. [30]. In brief, the outer primers, forward primer 5′-CCTCATTTTGCGGGTCACC-3′, and reverse primer 5′-TTTGACATACTTTCCAATCAAT-3′ were used. The thermocycling condition for the first round of PCR was performed at 94 °C for 3 min; 35 cycles of 94 °C for 1 min; 47 °C for 1 min; 72 °C for 1 min 30 s; and a final extension at 72 °C for 10 min. For the second round of PCR, the inner primers, forward primer 5′-TCACCATATTCTTGG GAAC-3′, and reverse primer 5′-AGGGTTTAAATGTATACCCA-3′ were used. The thermocycling condition was as follows: 94 °C for 3 min prior to 35 cycles of 94 °C for 1 min, 49 °C for 1 min, 72 °C for 1 min 15 s, followed by a final extension at 72 °C for 10 min. Nested PCR generated a PCR product size of 1206 bp. The positive samples were further amplified by a specific set of primers for full-length viral genome using the primer sets reported by Tangkijvanich [31] and Chook [32] to identify the core gene (Appendix A). Amplification of the DNA depended on the protocol that was set for the specific primers and reagents. However, multiple protocols and reagents were used to amplify each genome segment. Then, the PCR product was recovered and purified from agarose gel using a GenepHlow™ Gel/PCR Kit (Geneaid Biotech Ltd.) according to manufacturer’s protocol.

*(3) Nucleotide sequencing:* All positive PCR products were sequenced by the Sanger technique using a BigDye^®^ Terminator v3.1 Cycle Sequencing Kit and an Applied Biosystems^®^ 3730XL Genetic Analyzer (Thermo Fisher Scientifc Inc., Waltham, Massachusetts, USA). The partial nucleotide sequences of the S gene of HBV strains detected in this study were deposited in the Genbank database under accession numbers MK616477-MK616519 and OR061465-OR061470. (We chose the S gene and core gene to analyze the point mutations because previous reports demonstrated that mutations that are located in these genes may have a significant effect on disease severity and transmission effect).

*(4) Real-time polymerase chain reaction (real-time PCR):* The viral load of maternal samples was quantified by real-time PCR using reagents and specific primers, which were designed according to the sequences that were previously reported. Testing of the reagents and designed primers with the standard kit of RealStar^®^ HBV PCR Kit 1.0 (altona Diagnostics GmbH) was carried out. Real-time PCR was conducted using an Applied Biosystems 7000 real-time PCR system (ABI PRISM^®^ Thermo Fisher Scientific Inc., Waltham, MA, USA). Negative control wells and four quantification standards with ten-fold serial dilution (10^4^–10^1^ IU/µL) were included in each run. The real-time PCR was run for 45 cycles and the sigmoidal curve was assessed for detection of HBV-specific DNA and internal control. After finishing each run, baseline (threshold Ct) values were adjusted manually. Quantitation of each sample was achieved based on the standard curves. For cord blood and placental samples, real-time PCR results were categorized qualitatively into positive and negative groups based on whether the Ct values were lower or higher than the negative control wells, respectively, and the plus–minus function was used for confirmation. Real-time PCR had a lower limit of detection at 34 copies/mL (1.53 log copies/mL).

### 2.3. Molecular Analysis

*(1) Preparation of DNA sequencing for analysis:* Before molecular analysis, the DNA sequences were reviewed to ensure the quality of the sequences. All the processes, from amplification to purification, were repeated for low-quality DNA sequences. All HBV DNA strains were checked to ensure the correct nucleotide of each codon.

*(2) BLASTn search:* Each DNA sequence was added to MEGA-X software [33], an integrated tool for conducting automatic and manual sequence alignment, BLASTn search, percent of identity, and constructing phylogenetic tree. Further molecular analysis was done after adding DNA sequences to this program. The nucleotide sequences of the S gene of HBV strains detected in this study were used as query sequences to search for the most closely related sequence using the Nucleotide Basic Local Alignment Search Tool (BLASTn) search. 

*(3) Evaluation of the identity:* The percent identity between each HBV strain and the most closely related strain was noted in one of the columns shown on the BLASTn search page. The identities between the strains or with the other groups of reference strains were assigned using the function of the sequence identity matrix provided by the program. 

*(4) Point mutation:* After all of the strains were added to the MEGA-X program, the sequences were aligned with each other sequences and with the reference strains of the whole genome of HBV. We decided to use HBV subgenotype C1 because the similarity of the sequences would facilitate their alignments. The whole genome of C1 reference strains was used as the landmark of nucleotide sites to identify the mutation. The point mutations were analyzed point by point. Each point of the DNA sequence was analyzed in comparison with the reference. The lower left side of the figure shows the “site”/3215. This labels the site of the codon where the cursor was located along the total length of the 3215 codons of the whole HBV genome. Therefore, we can specify the point mutation by this function.

*(5) Construction of phylogenetic tree:* A phylogenetic tree was constructed based on partial nucleotide sequences of the small surface protein (S) gene of all matched HBV strains from maternal blood, placenta, and cord blood. The tree was generated using the neighbor-joining method in MEGA version X. The phylogenetic distance was estimated using the K2+G model. The S gene of HBV strains were compared to each other along with the most closely related sequence by a BLASTn server. 

*(6) Genotype and subgenotypes assignment:* We assigned the genotypes and subgenotypes of our strains using both of the most similar sequences that were previously reported and by analysis of the phylogenetic tree. The HBV strains detected in this study were further characterized by nucleotide sequence analysis in comparison with the prototype reference strains obtained from the GenBank database. For the most similar sequences, we accessed the information of 3–5 percent higher identity sequences to see the assigned genotype/subgenotype. Also, the phylogenetic tree was constructed. Analysis of the ancestor of each branch to assign the genotype and subgenotype was also conducted. Moreover, a comparison with the assigned genotype/subgenotype of the most closely related sequences from the phylogenetic tree was carried out to confirm the genotype/subgenotype of each strain.

### 2.4. Statistical Analysis

Statistical analysis was performed using the statistical software SPSS version 21.0 (IBM Corp. Released 2012; IBM SPSS Statistics for Windows, Version 21.0. Armonk, NY, USA). Percentages of cases with positivity for HBV DNA in the placenta and cord blood by both real-time PCR and nested PCR were calculated and presented. The ratio of each point mutations between the presence and absence of HBV DNA in placental tissue were compared. Relative risks (95% CI) of the genotypes on placental and cord blood HBV DNA detection were also calculated.

## 3. Results

The samples included in this study were 145 maternal blood, 142 placental tissue, and 142 cord blood samples of HBV-infected pregnant women. The HBV DNA detection rate by real-time PCR was different from that by nested PCR in all three sample types. The detection rate of HBV DNA in maternal blood was 83% (83 of 100) by real-time PCR, whereas it was 22% (32 of 145) by nested PCR. Similarly, the detection rate in placental tissue was 45.8% (44 of 96) by real-time PCR and 14.3% (20 of 140) by nested PCR. In addition, the detection rate in cord blood was 25.3% (24 of 95) by real-time PCR and 0.7% (1 of 139) by nested PCR, as presented in Table 1. The data suggest that real-time PCR is much more sensitive than nested PCR.

The distribution of HBV DNA-positive samples by real-time PCR is presented in Table 2.

About half of the cases that were HBV DNA-positive in the maternal blood were also HBV DNA-positive in the placenta (48 of 83; 57.8%).Nearly a quarter of cases were HBV DNA-positive in the cord blood but HBV DNA-negative in the placenta (5 of 22; 22.7%).A small number of cases were HBV DNA-positive in the placenta but HBV DNA-negative in the maternal blood (2 of 17; 11.7%).All cases that were HBV DNA-positive in the cord blood were also HBV DNA-positive in the maternal blood.

The distribution of HBV DNA-positive samples by nested PCR is shown in Table 3.

About half of the cases that were HBV DNA-positive in the maternal blood were also HBV DNA-positive in the placenta (6 of 13; 46.2%)Only one case was positive for HBV DNA in the cord blood, and this case was also positive for HBV DNA in the maternal blood and placenta.Showing a higher prevalence than that of real-time PCR, nested PCR detected 4 of 10 (40%) cases that were HBV DNA-positive in the placenta but HBV DNA-negative in the maternal blood.

A total of 145 maternal blood samples of HBsAg-positive pregnant women were tested by real-time PCR and nested PCR in order to determine the viral loads and HBV genotypes; 31 were positive for HBV DNA based on nested PCR amplification. The viral loads of cord blood samples were also quantitated by real-time PCR. The viral loads of maternal and cord blood samples and HBV genotypes detected in this study are shown in Table 4. The viral loads in 25 maternal blood samples varied from 2302 to 624,155,551 copies/mL, whereas the viral loads detected in 24 cord blood samples varied from 36 to 2,207,218 copies/mL. It should be noted that the viral loads could be quantitated only in 14 of 24 cord blood samples. The detected HBV strains included subgenotype C1 (77.4%; 24/31), subgenotype B9 (9.7%; 3/31), subgenotype C2 (3.2%; 1/31), subgenotype B2 (3.2%; 1/31), subgenotype B4 (3.2%; 1/31), and presumptive recombinant B4/ C2 subgenotype (3.2%; 1/31). The subgenotypes of HBV strains in the maternal blood, placental tissue, and cord blood are shown in Table 4.

To identify the association between genotype and the presence of HBV DNA in the placenta and cord blood, we considered all the subgenotypes as one prognostic factor because of their small number of positive cases, except subgenotype C1.

In this study, intrauterine HBV DNA detection was defined as the presence of HBV DNA in the placenta or cord blood. Accordingly, the results of the possible association between each genotype and intrauterine HBV DNA detection are separately presented in Table 5, as percentages and relative risks of placental and fetal (cord blood) HBV DNA detections for each genotype based on real-time PCR and nested PCR. Note that all cases of genotype B strains (4 of 4; 100%) had placental HBV DNA detection based on a positive test by real-time PCR in contrast to 71.4% of genotype C. Based on testing by both real-time PCR and nested PCR, both placental and fetal HBV DNA detections were more prevalent in genotype B; however, no statistically significant difference was found between both surrogate markers. 

The specific point mutations of the HBV genome were compared between positive and negative placental HBV DNA detection. To compare the significance of the mutation, we chose placental HBV DNA detection by real-time PCR as a surrogate marker for intrauterine HBV DNA detection. All points of mutation that were detected in our study were compared with previous reports as shown in Table 6. It should be noted that all point mutations observed in this study were not genotype-matched.

There were 52 point mutations detected in maternal blood sequences as shown in Table 6. Of these, 50 point mutations occurred in S ORF, and 2 point mutations occurred in X ORF. Twelve of these mutations were synonymous mutations (silent mutation) and forty were non-synonymous mutations. These point mutations randomly occurred, encompassing the whole genome. However, the mutations had no effect on placental HBV DNA detection.

No specific point mutation showing a significant effect on placental HBV DNA detection was identified in this study. However, based on the observed trends, the likely effect of each point mutation on placental HBV DNA detection were as follows:The point mutations that could be associated with more placental HBV DNA detection: C96A, G162A, C165T, A167G, T176C, G225A, G287A, A293G, C294T, C300T, T312C, C321A, C324T, A330G, C343T, C345G, A355G, T408G, C454T, C482A, A491T, G508C, A519G, G520A, T562A, T581A, T592C, G633A, T636A, A667T, and T724CThe point mutations that could be associated with less placental HBV DNA detection: A162G, T213C, G285A, C339A, G348A, G390A, T400A, T438G, A453G, T531G, T531C, T562G, and C720T

The phylogenetic tree was constructed to demonstrate the identity between maternal, placental, and fetal HBV sequences, along with reference strains, as shown in Figure 1.

From phylogenetic analysis, HBV DNA sequences of maternal blood, placenta, and cord blood of each case were identical. 

## 4. Discussion

This study investigated the presence of HBV DNA in maternal blood, the placenta, and fetal blood and did not directly investigate placental/fetal HBV infection. Nevertheless, the presence of HBV DNA in the placenta and fetus might imply placental and fetal exposure to HBV. Our evidence may guide or alert HBV researchers to new insights into the potential effects of viral genotypes and intrauterine HBV DNA detection. Insights gained from this study are as follows: (1) Fetal HBV DNA detection was identified in approximately 25% of HBV-positive mothers in association with the presence of HBV in the placenta in most cases; (2) We demonstrated that HBV DNA could be detected in the placenta with high prevalence; (3) HBV genotype B had a higher tendency to enhance intrauterine HBV DNA detection than HBV genotype C. The C454T mutation had a tendency to be associated with more placental HBV DNA detection rate, but that of T400A tended to have a lower detection rate; (4) Mutation was unlikely to occur during intrauterine exposure because HBV DNA strains in the maternal–placental–cord blood collected from the same cases were identical. 

### 4.1. New Insights Regarding a Relationship between the HBV Genotypes and Genetic Variability with the Risk of Intrau-Terine HBV DNA Detection

First, all the samples were tested by nested PCR and real-time PCR to evaluate the detection rate of each technique for each sample type. We found that real-time PCR is much more sensitive than nested PCR. The surrogate markers of the presence of HBV DNA in the placenta and cord blood were also tested, and the results indicate that HBV DNA detection in the placenta is more prevalent than in cord blood. Our previous report indicated that fetal HBV DNA detection may be mainly exposed in utero rather than fetal contamination during delivery; we found that fetal HBV DNA detection is strongly associated with placental HBV DNA detection (78.3%) [43]. From this assumption, the presence of HBV DNA in fetal cord blood likely occurs during the antepartum period as evidence that HBV is accumulated in the placenta and might be then transferred as HBV to the fetus.

The main pattern of HBV distribution based on HBV DNA detection using real-time PCR and nested PCR suggested that the maternal blood would be infected first, followed by HBV accumulation in the placenta, and lastly in the cord blood. However, the distribution of HBV DNA detection in maternal blood, placental tissue, and cord blood in some cases was inconsistent. Interestingly, we found cases of HBV DNA detected in the placenta, but we did not detect HBV DNA in the maternal blood. Thus, the placenta may be one of the organs in which HBV can accumulate, and it can serve as a reservoir of organisms escaping from maternal immunity and treatment. However, we found positive cases of HBV DNA in the cord blood without placental HBV DNA detection, indicating that, in some cases, HBV could possibly pass to the fetus without accumulation in the placenta.

The prevalence of various HBV genotypes and subgenotypes varies in different geographical areas. According to the findings of our study (submitted for publication), the HBV subgenotype C1 is the most common subgenotype circulating in pregnant women in Northern Thailand and accounts for 77.4% of cases, followed by subgenotypes B9 (9.7%), C2 (3.2%), B2 (3.2%), B4 (3.2%), and presumptive recombinant B4/C2 subgenotype (3.2%). Due to their small number of positive cases, we considered all the subgenotypes as one prognostic factor, except subgenotype C1. The association between genotype (B and C) and the presence of HBV DNA in the placenta and cord blood (based on real-time PCR and nested PCR) was evaluated, and it was demonstrated that genotype B showed a significantly higher risk of placental HBV DNA detection rate than genotype C. The relative risks (95% CI) were 1.40 (1.07–1.84), 1.14 (0.46–2.84), and 1.36 (0.68–2.72), and 10.71 (0.49–235.23) for placental HBV DNA detection and cord blood HBV DNA detection based on real-time PCR and nested PCR, respectively. Genotype B showed a higher prevalence of intrauterine HBV DNA detection than genotype C; however, no statistically significant difference was found between both surrogate markers with both methods of detection. To the best of our knowledge, no previous study on the association between genotype and intrauterine HBV DNA detection has been conducted; we are the first to report that genotype B could be a more appropriate prognostic marker for intrauterine HBV DNA detection compared with genotype C. To obtain additional information on the issue, further studies are needed to compare the risk of intrauterine HBV DNA detection between genotypes B and C, which are the major genotypes circulating in our geographical area. Also, studies on other genotypes in other geographical areas are needed.

This study also evaluated specific point mutations of the HBV genome and placental HBV DNA detection based on HBV DNA positivity using real-time PCR. The results show that no specific point mutation had a significant effect on placental HBV DNA detection; however, the trend in each mutation regarding placental HBV DNA detection was suggested. 

We found that the point mutation positions that could be associated with more placental HBV DNA detection were C96A, G162A, C165T, A167G, T176C, G225A, G287A, A293G, C294T, C300T, T312C, C321A, C324T, A330G, C343T, C345G, A355G, T408G, C454T, C482A, A491T, G508C, A519G, G520A, T562A, T581A, T592C, G633A, T636A, A667T, and T724C. On the other hand, the point mutations that could be associated with less placental HBV DNA detection were A162G, T213C, G285A, C339A, G348A, G390A, T400A, T438G, A453G, T531G, T531C, T562G, and C720T. Most of these point mutations were not previously reported; however, some point mutations have been reported. The effect of the enhancement of the presence or absence of HBV DNA in the placenta by each point mutation is closely related to the results of previous reports regarding potential effects on intrauterine transmission [34,35,36,37,38,39,40,41,42]. The mutations at G162A, A167G, and A667T showed a trend in increasing intrauterine HBV DNA detection, which agrees with previous reports, suggesting that these mutations could enhance HBV DNA detection. On the contrary, the mutations at A162G and T531G were found more in the group of no HBV DNA detection in the placenta. This effect is in accordance with previous reports [34,35,36,37,38,39,40,41,42].

The most interesting mutations from this study are the C454T and T400A mutations. The C454T mutation was present in the group with HBV DNA detection in the placenta in 7 of 19 cases (36.8%), but this type of mutation was not observed in the group in which HBV DNA was not detected in the placenta with the highest relative risk of 5.25 (0.34–80.52). This mutation may be a potential prognostic factor that enhances intrauterine HBV DNA detection. On the other hand, the T400A mutation may have a protective effect. This mutation was found in two of six cases (33.3%) in the group of no HBV DNA detection in the placenta, but it was not detected in the group in which HBV DNA was detected in the placenta with a relative risk of 0.07 (0.01–1.29), although this was not statistically significant. It should be pointed out that no previous study has reported these mutations as prognostic factors.

The next interesting point is related to the T562A and T562G mutations, which have the same site of mutation but different nucleotide substitution, showing opposite effects. The T562A mutation tended to have more intrauterine HBV DNA detection (relative risk of 3.15; 95% CI 0.19–51.40) in contrast to the T562G mutation, which tended to have less intrauterine HBV DNA detection (relative risk of 0.12; 95% CI 0.01–2.55).

Lastly, our findings show that no mutation occurred during the intrauterine HBV exposure process. This negative finding is also important to document since it can provide reassurance that the effectiveness of antiviral drug treatment in the mothers can also be reproduced in the fetus.

However, many studies reported some specific point mutations in the pre-S and S regions of the HBV genome that could be significant factors of vertical transmission. That being said, the results are inconsistent due to the different genotypes of HBV and types of clinical samples included in each study. To the best of our knowledge, point mutations of C454T, T400A, T562A, and T562G, all of which are coded on the S gene in the HBV genome, likely play a potential role in intrauterine HBV DNA detection, although the changes secondary to these mutations are still unclear.

### 4.2. Additional Interesting Findings Regarding Intrauterine HBV Exposure

There was a significant number of cases with negative HBV DNA detection in the maternal blood but positive HBV DNA detection in the placenta. This finding implies that the amount of HBV DNA in the placenta was higher than that in the maternal blood, suggesting that the placenta could possibly be a site where HBV can be accumulated. Accordingly, the placenta can be a reservoir of HBV, thus readily facilitating fetal HBV DNA detection. Nevertheless, nearly a quarter of cases that were HBV DNA-positive in the cord blood exhibited no HBV DNA detection in the placenta (5 of 22; 22.7%). This finding indicates that there were a significant number of cases with the presence of HBV DNA in the fetus but absence in the placenta. This finding partly supports the hypothesis that the presence of HBV DNA in fetuses could result from direct contamination of maternal blood from the intervillous space to fetal blood in the villous vessels without significant HBV DNA detection in the placenta. It is noteworthy that most cases of fetal HBV DNA detection had placental HBV DNA detection and most cases of placental HBV DNA detection had maternal blood HBV DNA detection (significant association). It can reasonably be assumed that intrauterine HBV exposure had already occurred in utero since the detection of HBV DNA in placenta is certainly an intrauterine and not perinatal event. This statement is supported by the fact that nearly all cases of maternal HBV DNA detection without HBV DNA detection in the placenta had no cord blood HBV DNA detection.

The results of this study indirectly support two mechanisms of fetal HBV exposure, mainly transferring in utero through the placenta and through direct maternal–fetal blood contamination. Most cases with cord blood HBV DNA detected immediately after birth had evidence of placental HBV DNA, suggesting intrauterine HBV exposure. It is possible that such fetuses had already been exposed to HBV in utero and were likely associated, at least in part, with failure of immunoprophylaxis after birth. Cord blood HBV DNA detection could be categorized into two groups: with and without evidence of placental HBV DNA detection. The latter group is likely caused by direct maternal–fetal contamination during delivery. Note that this study does not indicate that intrauterine infection is the most common mechanism of fetal infection, but cord blood HBV DNA detected immediately after birth is likely caused by prenatal intrauterine HBV exposure rather than contamination of maternal blood during delivery. 

The weaknesses of this study are as follows: (1) The sample size is relatively small and further studies are encouraged to confirm the new prognostic factors of both genotypes and point mutations; (2) viral load measurements and HBeAg testing were not incorporated in the analysis; (3) notably, the detection of HBV DNA in the placenta or in the fetus might not represent intrauterine transmission. Therefore, the association of the presence of HBV DNA and genotypes in the mothers, placentas, and fetuses observed in this study might simply be artefactual and not a mechanism of transmission, particularly given the lack of any observations that mothers with viral load < 20,000 IU/mL can transmit to their newborns if given prompt vaccination or HBIG; (4) We use the point mutations in the S gene as a surrogate marker of intrauterine HBV DNA detection because the previous reports demonstrated that mutations located in the S gene have a significant effect on disease severity and HBV transmission [34,35,36,37,38,44,45]. However, the occurrence of intrauterine infection has not been definitively shown in this study. Nevertheless, it is reasonable to make an assumption that the presence of HBV DNA in the fetus may indirectly imply viral accumulation in the placenta and transfer to the fetus, although this assumption needs further study to be confirmed.

Suggested future studies are as follows: (1) studies to determine whether a combination of genotype, viral load, or positivity of HBeAg can improve the accuracy in predicting vertical transmission; (2) follow-up of the newborns in this cohort to identify the relationship between cases with placental HBV DNA detection and immunoprophylaxis failure; and (3) whether antiviral drug during pregnancy can effectively reduce the rate of placental and fetal HBV DNA detection should be explored.

## 5. Conclusions

The main evidence provided by this study is that genotypes B had a higher trend in the detection rate of HBV DNA in the placenta and fetus than genotype C. Though the detection rate of HBV DNA might not directly represent placental/fetal infection, it is reasonable to hypothesize that the genotypes might have a potential effect on intrauterine HBV DNA detection, encouraging future studies to elucidate the effect of genotypes on intrauterine infection. Additionally, the C454T mutation showed a tendency to have a higher HBV DNA detection rate in the placenta, but the T400A mutation tends to have a lower detection rate. Furthermore, mutation is unlikely to occur during the intrauterine HBV exposure process. 

## Figures and Tables

**Figure 1 viruses-15-01729-f001:**
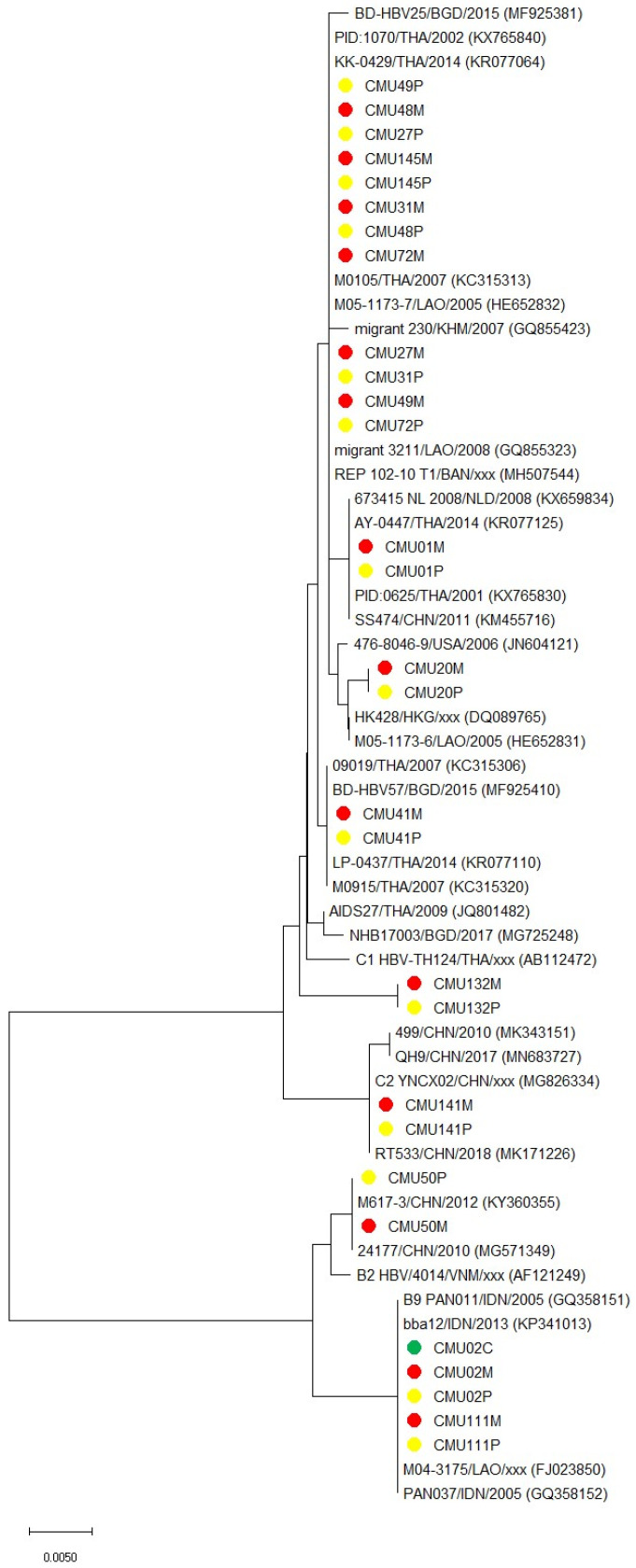
Phylogenetic tree based on partial nucleotide sequences of the small surface protein (S) gene of all matched HBV strains from maternal blood, placenta, and cord blood. The tree was generated using the neighbor-joining method in MEGA version X. The phylogenetic distance was estimated using the K2+G model. The S genes of HBV strains were compared to each other, along with the most closely related sequence by a BLASTn server. The HBV strains detected in the present study are indicated by a circle: red circle for maternal blood, yellow circle for placenta, and green circle for cord blood. The scale bar indicates the branch length for a 0.005-nucleotide difference.

**Table 1 viruses-15-01729-t001:** Detection rates of HBV DNA by real-time PCR and nested PCR.

Type of Samples	Positive Real-Time PCR (%)	Positive Nested PCR (%)
Maternal blood	83/100 (83%)	32/145 (22%)
Placental tissue	44/96 (45.8%)	20/140 (14.3%)
Cord blood	24/95 (25.3%)	1/139 (0.7%)

**Table 2 viruses-15-01729-t002:** Distribution of positive HBV DNA samples by real-time PCR (M: maternal blood; P: placenta; C: cord blood).

No.	M	P	C	No.	M	P	C	No.	M	P	C	No.	M	P	C
1				26				51				76			
2				27				52				77			
3				28				53				78			
4				29				54				79			
5				30				55				80			
6				31				56				81			
7				32				57				82			
8				33				58				83			
9				34				59				84			
10				35				60				85			
11				36				61				86			
12				37				62				87			
13				38				63				88			
14				39				64				89			
15				40				65				90			
16				41				66				91			
17				42				67				92			
18				43				68				93			
19				44				69				94			
20				45				70				95			
21				46				71				96			
22				47				72				97			
23				48				73				98			
24				49				74				99			
25				50				75				100			

**Table 3 viruses-15-01729-t003:** Distribution of HBV DNA-positive samples by nested PCR (M: maternal blood; P: placenta; C: cord blood).

No.	M	P	C	No.	M	P	C	No.	M	P	C	No.	M	P	C	No.	M	P	C
1				31				61				91				121			
2				32				62				92				122			
3				33				63				93				123			
4				34				64				94				124			
5				35				65				95				125			
6				36				66				96				126			
7				37				67				97				127			
8				38				68				98				128			
9				39				69				99				129			
10				40				70				100				130			
11				41				71				101				131			
12				42				72				102				132			
13				43				73				103				133			
14				44				74				104				134			
15				45				75				105				135			
16				46				76				106				136			
17				47				77				107				137			
18				48				78				108				138			
19				49				79				109				139			
20				50				80				110				140			
21				51				81				111				141			
22				52				82				112				142			
23				53				83				113				143			
24				54				84				114				144			
25				55				85				115				145			
26				56				86				116							
27				57				87				117							
28				58				88				118							
29				59				89				119							
30				60				90				120							

**Table 4 viruses-15-01729-t004:** Genotypes/subgenotypes and viral loads of HBV strains in the maternal blood, placental tissue, and cord blood tested by real-time PCR and nested PCR.

HBV Strains	Viral Load (Copies/mL)	Genotypes/Subgenotypes	HBV DNA Detected in Placental Tissue		HBV DNA Detected in Cord Blood		
			Real-Time PCR	Nested PCR	Real-Time PCR	Viral Load (Copies/mL)	Nested PCR
CMU01M	2302	C1	positive	positive	negative	-	negative
CMU02M	411,679,497	B9	positive	positive	positive	249,805	positive
CMU07M	3412	C1	negative	negative	positive	180	negative
CMU11M	31,558	C1	negative	negative	negative	-	negative
CMU18M	44,506	C1	negative	negative	negative	-	negative
CMU20M	98,774	C1	positive	positive	negative	-	negative
CMU25M	659,452,481	C1	positive	negative	positive	81	negative
CMU27M	438,501,710	C1	positive	positive	positive	1584	negative
CMU28M	32,707	C1	negative	negative	negative	-	negative
CMU31M	226,080,088	C1	positive	positive	positive	271	negative
CMU35M	13,051,476	C1	positive	negative	negative	-	negative
CMU40M	120,425	C1	positive	negative	positive	2639	negative
CMU41M	624,155,551	C1	positive	positive	positive	2,207,218	negative
CMU44M	1184	C1	positive	negative	negative	-	negative
CMU48M	207,070	C1	positive	positive	positive	412	negative
CMU49M	72,127	C1	positive	positive	positive	48	negative
CMU50M	353,604,282	B2	positive	positive	positive	226	negative
CMU56M	8182	C1	negative	negative	negative	-	negative
CMU57M	11,296	C1	negative	negative	negative	-	negative
CMU69M	346,889	B **** B4/C2?	positive	negative	positive	160	negative
CMU72M	609,448,400	C1	positive	positive	positive	36	negative
CMU80M	51,614	C1	positive	negative	ND	ND	ND
CMU87M	883,490	B9	positive	negative	negative	-	negative
CMU88M	163,773,228	C1	positive	negative	positive	154,075	negative
CMU95M	4529	C1	positive	negative	positive	1227	negative
CMU108M	ND	C1	ND	negative	ND	ND	negative
CMU111M	ND	B9	ND	positive	ND	ND	negative
CMU125M	ND	B4	ND	negative	ND	ND	negative
CMU132M	ND	C1	ND	positive	ND	ND	negative
CMU141M	ND	C2	ND	positive	ND	ND	negative
CMU145M	ND	C1	ND	positive	ND	ND	negative

ND = not done; the viral loads of case #CMU108M, CMU111M, CMU125M, CMU132M, CMU141M, CMU145M were not available.

**Table 5 viruses-15-01729-t005:** Percentage and relative risk of placental and fetal HBV DNA detections for each genotype based on a positive test by real-time PCR and nested PCR.

**Genotype**		**Presence of** **HBV DNA in Placenta (%)**	**Absence of** **HBV DNA in Placenta (%)**	**Total**	**Relative Risk (95% CI)**	* **p** * **-Value**
Real-time PCR	B	4 (100%)	0 (0%)	4	1.40 (1.07–1.84)	0.054
	C	15 (71.4%)	6 (28.6%)	21		
Nested PCR	B	3 (50%)	3 (50%)	6	1.14 (0.46–2.84)	0.791
	C	11 (44%)	14 (56%)	25		
**Genotype**		**Presence of** **HBV DNA in Cord Blood (%)**	**Absence of** **HBV DNA in Cord Blood (%)**	**Total**	**Relative Risk (95% CI)**	* **p** * **-Value**
Real-time PCR	B	3 (75%)	1 (25%)	4	1.36 (0.68–2.72)	0.615
	C	11 (55%)	9 (45%)	20		
Nested PCR	B	1 (16.7%)	5 (83.3%)	6	10.71 (0.49–235.23	0.200
	C	0 (0%)	24 (100%)	24		

**Table 6 viruses-15-01729-t006:** Specific point mutations of the HBV genome and placental HBV DNA detection by real-time PCR in comparison with the previous reports on potential HBV vertical transmission.

Mutation Point	Presence of HBV DNA in Placenta (Positive/Negative Mutation)	Absence of HBV DNA in Placenta (Positive/Negative Mutation)	Relative Risk (95% Confidence Interval)	Potential Effect on Placental HBV DNA Detection	Previous Reports on Potential Intrauterine Transmission
T31C	-	-			Enhance [34]
T52C	-	-			Enhance [34]
C96T	-	-			Enhance [35]
C96A	2/8	0/2	1.36 (0.08–21.44)	↑	-
G145A	-	-			Enhance [36]
G162A	4/15	0/6	3.15 (0.19–51.40)	↑↑	Enhance [35]
A162G	15/4	6/0	0.79 (0.63–1.00)	↓	Protect [37]
C165T	4/15	0/6	3.15 (0.19–51.40)	↑↑	-
A167G	4/15	0/6	3.15 (0.19–51.40)	↑↑	Enhance [38]
T176C	5/14	0/6	3.85 (0.24–61.09)	↑↑	-
T213C	0/19	2/4	0.07 (0.01–1.29)	↓↓↓	-
G225A	6/13	0/6	4.55 (0.29–70.80)	↑↑↑	-
G285A	1/18	1/5	0.32 (0.02–4.32)	↓	-
G287A	4/15	0/6	3.15 (0.19–51.40)	↑↑	-
A293G	4/15	0/6	3.15 (0.19–51.40)	↑↑	-
C294T	4/15	0/6	3.15 (0.19–51.40)	↑↑	-
C300T	4/15	0/6	3.15 (0.19–51.40)	↑↑	-
T312C	4/15	0/6	3.15 (0.19–51.40)	↑↑	-
C321A	4/15	0/6	3.15 (0.19–51.40)	↑↑	-
C324T	5/14	0/6	3.85 (0.24–61.09)	↑↑	-
A330G	4/15	0/6	3.15 (0.19–51.40)	↑↑	-
C339T	-	-			Protect [37]
C339A	0/19	1/5	0.12 (0.01–2.55)	↓	-
C343T	3/16	0/6	2.45 (0.14–41.74)	↑	-
C345G	4/15	0/6	3.15 (0.19–51.40)	↑↑	-
G348A	0 /19	1/5	0.12 (0.01–2.55)	↓	-
C354A	1/18	0/6	1.05 (0.05–22.91)	↔	-
A355G	3/16	0/6	2.45 (0.14–41.74)	↑	-
T357C	1/18	0/6	1.05 (0.05–22.91)	↔	-
G390A	0/19	1/5	0.12 (0.01–2.55)	↓	-
T400A *	0/19	2/4	0.07 (0.01–1.29)	↓↓	-
G403A	-	-			Enhance [38]
C407A	-	-			Enhance [38]
T408G	4/15	0/6	3.15 (0.19–51.40)	↑↑	-
T434C	-	-			Protect [37]
T438G	0/19	1/5	0.12 (0.01–2.55)	↓	-
T441C	1/18	0/6	1.05 (0.05–22.91)	↔	-
A453G	1/18	1/5	0.32 (0.02–4.32)	↓	-
C454T *	7/12	0/6	5.25 (0.34–80.52)	↑↑↑↑	-
T462C	-	-			Protect [37]
T473C	-	-			Enhance [35]
C482A	4/15	0/6	3.15 (0.19–51.40)	↑↑	-
A491T	5/14	0/6	3.85 (0.24–61.09)	↑↑	-
G508C	4/15	0/6	3.15 (0.19–51.40)	↑↑	-
A519G	3/16	0/6	2.45 (0.14–41.74)	↑	-
G520A	4/15	0/6	3.15 (0.19–51.40)	↑↑	-
T531G	0/19	1/5	0.12 (0.01–2.55)	↓	Protect [37]
T531C	5/14	3/3	0.53 (0.18–1.58)	↓	-
T562A *	4/15	0/6	3.15 (0.19–51.40)	↑↑	-
T562G *	0/19	1/5	0.12 (0.01–2.55)	↓	-
T581A	4/15	0/6	3.15 (0.19–51.40)	↑↑	-
T592C	4/15	0/6	3.15 (0.19–51.40)	↑↑	-
T598C	9/10	2/4	1.42 (0.42–4.85)	↔	-
G633A	4/15	0/6	3.15 (0.19–51.40)	↑↑	-
T636A	4/15	0/6	3.15 (0.19–51.40)	↑↑	-
A667T	5/14	0/6	3.85 (0.24–61.09)	↑↑	Enhance [38]
T670G	-	-			Enhance [38]
A673G	-	-			Enhance [38]
A680C	-	-			Enhance [38]
C705T	-	-			Enhance [35]
C720T	1/18	1/5	0.32 (0.02–4.32)	↓	-
T724C	4/15	0/6	3.15 (0.19–51.40)	↑↑	-
A753T	2/17	0/6	1.75 (0.10–32.18)	↔	-
A802G	-	-			Protect [37]
T810C	-	-			Protect [37]
G1719T	-	-			Enhance [35]
G1742T	-	-			Enhance [35]
A1762T	1/1	-			Enhance [35]
G1764A	1/1	-			Enhance [35]
A1762T/ G1764A	1/1	-			Enhance [35] Protect [39] Increase severity [40,41]
G1896A	-	-			Enhance [42] Protect [39] Increase severity [40,41]
1899-A	-	-			Increase severity [40,41]
C2875A	-	-			Protect [37]
C2990T	-	-			Protect [38]
C3000A	6/2	2/0	0.75 (0.50–1.12)	↔	Protect [37]
C3116T	-	-			Enhance [34,35]
C3175T	-	-			Enhance [35]
T3205A	-	-			Protect [38]
G3212A	-	-			Protect [37]

* There are four common silent mutations: C454T, T400A, T562A, and T562G.

## Data Availability

The datasets analyzed during the current study are available from the corresponding author upon reasonable request. The HBV DNA sequences were submitted to GenBank under accession number MK616477-MK616519 and OR061465-OR061470.

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
