# Peer review of "Possible Association between Genetic Diversity of Hepatitis B Virus and Its Effect on the Detection Rate of Hepatitis B Virus DNA in the Placenta and Fetus"

_viruses, 2023, doi:10.3390/v15081729_

Round 1

Reviewer 1 Report (Previous Reviewer 1)

1) The abstract is too long. It has 378 words. Most journals restrict the abstracts to 400 words or less. Some even have 200 words restrictrion. I am not sure about the policy of Viruses but I am quite certain it is about 300 words or less. This is to be confirmed by the editor. Keeping the abstract concise and short is to the advantage of the readers and writers better, not just the journals. Scientists and clinicians are often too busy to read the abtract carefully. They normally galnce through the abstract quickly to see if the article is relevant to their interest. If the abstract is too long, they may not even bother to read the abstract, not to mention the article.

2) The descriptions on the legends of the tables are too short.Again, many scientists and clinicians are too busy to read a paper thoroughly and therefore rely on the glance of the tables, figures and their legends, to gain a quick understanding of the paper. If there is no description on the legends, they may not read the paper any fhurther.

Author Response

Comments and Suggestions for Authors

1) The abstract is too long. It has 474 words. Most journals restrict the abstracts to 400 words or less. Some even have 200 words restrictrion. I am not sure about the policy of Viruses but I am quite certain it is about 300 words or less. This is to be confirmed by the editor. Keeping the abstract concise and short is to the advantage of the readers and writers better, not just the journals. Scientists and clinicians are often too busy to read the abtract carefully. They normally glance through the abstract quickly to see if the article is relevant to their interest. If the abstract is too long, they may not even bother to read the abstract, not to mention the article.

Response: The abstract word count is reduced from 474 to 360.

2) The descriptions on the legends of the tables are too short. Again, many scientists and clinicians are too busy to read a paper thoroughly and therefore rely on the glance of the tables, figures and their legends, to gain a quick understanding of the paper. If there is no description on the legends, they may not read the paper any further.

Response: All table heads are succinct but include the main concept for understanding at glance. Short description of each table is also provided in text just before the table. We make a small request to keep the way it is, to avoid redundant description. However, if the reviewer or the editors strongly recommend long description, we are willing to comply.

Reviewer 2 Report (Previous Reviewer 2)

All previous comments have now been addressed adequately.

Minor editing by the journal may still be needed.

Author Response

All previous comments have now been addressed adequately.

Response: Thank you.

Comments on the Quality of English Language

Minor editing by the journal may still be needed.

Response: English has been proof-edited by the English proofreading service.

This manuscript is a resubmission of an earlier submission. The following is a list of the peer review reports and author responses from that submission.

Round 1

Reviewer 1 Report

While the authors presented some interesting results in this paper much improvements should be made.

1) The Abstract is tool long. Noramlly most journals allow word limit of 200-350 words but this is about 520 words. There is an important reason that journals have word limits to the abstract. Scientists usually prefer to take a quick glance at the abstract before deciding if the paper is interesting enough for them to rea. If the abstract is too long, they may not even bother to read the abstract, not to mention the paper.

2) There is no description of the kind of virus HBV is. eg RNA/DMA, family, genera.. This is important as it would have implication on how the virus will behave and what kind of proteins it has.

3) There is no description of the viral proteins or the virion physiology of HBV. This is important as we will see later.  The authors can get information pertaining to these at:

https://www.ncbi.nlm.nih.gov/pmc/articles/PMC2809016/

3) There is a very important reason  that the authors must talk about proteins and virion. The viral proteins are responsible for all the behaviors of all viruses and organisms including intrauterine transmission of HBV. The authors described genetic mutations that aprovide for greater intrauterine transmission. But which viral protein(s) is affected by the mutation? Is it the surface protein, which provides for easier viral entry? Or is it some other proteins?  What are the functions of the proteins involved?

An example is the following paper:

https://pubmed.ncbi.nlm.nih.gov/31698857/

The paper describes how Zika and yeloow fever viruses are able to enter vital organs such as the brain and placenta because of mutations (disorder) at the outer shells of the viruses.

While I understand that many of what I am saying may be beyond the scope of this paper, the authors should attempt to discuss such matters as other scientists are likely to investigate into this matter using the data presented.

4) The paper mentioned "Correlation" but technically we cannot mention correaltion unless we can show r (correlation coefficient) >= 0.5 or coefficient of determination (r2) >= 0.25. If you are unable to show these, then you can try to show "relationship" or "link" but you need to show statistical significance (p=< 0.5) but I am not able to see this. There is no mentoned of the statistical model used (e.g mutivariate analysis, regression )

Author Response

Reviewer Comments

Reviewer 1 (highlighted in red)

While the authors presented some interesting results in this paper much improvements should be made.

1) The Abstract is too long. Normally most journals allow word limit of 200-350 words but this is about 520 words. There is an important reason that journals have word limits to the abstract. Scientists usually prefer to take a quick glance at the abstract before deciding if the paper is interesting enough for them to rea. If the abstract is too long, they may not even bother to read the abstract, not to mention the paper.

Response 1: In revised MS, the abstract is shortened to have word count of 349.

2) There is no description of the kind of virus HBV is. eg RNA/DMA, family, genera.. This is important as it would have implication on how the virus will behave and what kind of proteins it has.

Response 2: In revised MS, the description is added in the first paragraph of “Introduction” section. Please see page 2 lines 42-57.

3) There is no description of the viral proteins or the virion physiology of HBV. This is important as we will see later.  The authors can get information pertaining to these at:

https://www.ncbi.nlm.nih.gov/pmc/articles/PMC2809016/

Response 3: In revised MS, the description is added in the first paragraph of “Introduction” section. Please see page 2 lines 47-54.

4) There is a very important reason that the authors must talk about proteins and virion. The viral proteins are responsible for all the behaviors of all viruses and organisms including intrauterine transmission of HBV. The authors described genetic mutations that provide for greater intrauterine transmission. But which viral protein(s) is affected by the mutation? Is it the surface protein, which provides for easier viral entry? Or is it some other proteins?  What are the functions of the proteins involved?

An example is the following paper:

https://pubmed.ncbi.nlm.nih.gov/31698857/

The paper describes how Zika and yellow fever viruses are able to enter vital organs such as the brain and placenta because of mutations (disorder) at the outer shells of the viruses.

While I understand that many of what I am saying may be beyond the scope of this paper, the authors should attempt to discuss such matters as other scientists are likely to investigate into this matter using the data presented.

Response 4: In revised MS, we discuss about this in “Discussion” section. Please see page 16 lines 422-427, as highlighted in red.

5) The paper mentioned "Correlation" but technically we cannot mention correaltion unless we can show r (correlation coefficient) >= 0.5 or coefficient of determination (r2) >= 0.25. If you are unable to show these, then you can try to show "relationship" or "link" but you need to show statistical significance (p=< 0.5) but I am not able to see this. There is no mentoned of the statistical model used (e.g mutivariate analysis, regression )

Response 5: In revised MS, we change “correlation” to “association” or “relationship” throughout the MS, and p-values are added as presented in Table 5. However, the significance of genotype transmission and intrauterine infection are contradictory, p-value >0.05 implying not significant while relative risk of not including 1 implying significance. This might be caused by a relatively small sample size. Thus, we conclude that there is a trend of an increased risk of intrauterine infection, rather than significant increase.

Reviewer 2 Report

This is a cross-section study of intrauterine transmission of HBV and viral diversity in HBsAg-positive women in Thailand.

The population size of 145 pregnant women with HBV is quite reasonable.  The methods are well described and appropriate for a study of this nature.  However, many of the results are poorly described and require significant improvement.

Moreover, there are several grammatical errors and awkward phrases throughout the manuscript.  The revised manuscript should be reviewed carefully by a native English speaker and/or a professional editing service.

The introduction should include information about HBV treatments available in Thailand, as well as when HBV vaccination is offered and how often it is administered at the appropriate time points in infants.

What is the lower limit of detection of the real-time PCR for HBV DNA that was used?

Are the S gene primers described in section 2.2 and reference 24 able to amplify all HBV genotypes equally?  If not, this would greatly influence the findings – and interpretation – presented in the current study.

In section 2.3, how was “high-quality DNA” defined / determined?

In section 2.3, why was the construction of the phylogenetic tree published separately?  That information should be included here as well.

In tables 5-8, does “non-infection” refer to “no placental infection”?

Tables 5 and 6 can be combined, as can tables 7 and 8.

The identification of point mutations is poorly described.  Is one maternal study sequence compared to one reference sequence or to many references or to the corresponding infant sequence?

Table 9 shows nucleotide variations.  How many of these mutations lead to amino acid changes?  Synonymous versus non-synonymous changes should also be discussed in the discussion when point mutations are listed.

Are the data in Table 9 matched by genotype?  It would be inappropriate to compared across genotypes as this may identify genotype-specific mutations rather than mutations associated with intrauterine transmission.

The statements that “mutation is unlikely to occur during intrauterine infection” and’ . . . no mutation occurred during intrauterine infection process” in the Discussion contradict the rest of the paper . . . particularly Table 9 in which multiple potential mutations are reported.

In the discussion section, the authors must reference other studies that have evaluated HBV DNA presence/absence in the placenta.  Are the findings reported here supported by other publications?

The data presented in supplementary figure 3 are not described adequately in the text.  How similar are HBV sequences from the same genotype?

Is there sequence data available from the corresponding infants?  If not, how was transmission to the infant documented / confirmed?

Supplementary figure 6 includes only 6 study sequences.  Where are the remaining ones?

There are several grammatical errors and awkward phrases throughout the manuscript.  The revised manuscript should be reviewed carefully by a native English speaker and/or a professional editing service.

Author Response

Reviewer Comments

Reviewer 2 (highlighted in blue)

Comments and Suggestions for Authors

This is a cross-section study of intrauterine transmission of HBV and viral diversity in HBsAg-positive women in Thailand.

The population size of 145 pregnant women with HBV is quite reasonable.  The methods are well described and appropriate for a study of this nature.  However, many of the results are poorly described and require significant improvement.

Moreover, there are several grammatical errors and awkward phrases throughout the manuscript.  The revised manuscript should be reviewed carefully by a native English speaker and/or a professional editing service.

Response 1: The revised MS has been checked and edited by the native English proofreader, please see the attached certificate.

The introduction should include information about HBV treatments available in Thailand, as well as when HBV vaccination is offered and how often it is administered at the appropriate time points in infants.

Response 2: In revised MS, information about treatment and vaccination in Thailand are included as suggested, as highlighted at the end of the third paragraph of “Introduction” section. Please see pages 2-3 lines 91-95.

What is the lower limit of detection of the real-time PCR for HBV DNA that was used?

Response 3: In revised MS, the lower limit is added in the “Methods” section. Please see page 5 lines 200-201, as highlighted at the end of paragraph item “4) Real-time polymerase chain reaction (real-time PCR)” of subheading 2.2.

Are the S gene primers described in section 2.2 and reference 24 able to amplify all HBV genotypes equally? If not, this would greatly influence the findings – and interpretation – presented in the current study.

Response 4: Yes, the S gene primers are able to amplify all HBV genotypes equally. Also associated with viral load detected by real time PCR.

In section 2.3, how was “high-quality DNA” defined / determined?

Response 5: We want to present that DNA sequences included in analysis  are reliable, they must be clear-cut, no overlapping peaks. However, the preceding sentence is “… was reviewed to ensure quality.” So in revised MS, we delete the sentence “Only high-quality DNA sequences were used for analysis” to avoid confusion and repetition.

In section 2.3, why was the construction of the phylogenetic tree published separately?  That information should be included here as well.

Response 6: Our project consists of two parts, Part I: epidemiology of HBV genotypes in pregnant women in Thailand that require the construction of phylogenetic tree and Part II: effects of genotypes on intrauterine infection. Part I has already been submitted separately, and under review by other journal. Because of different purpose and the combination of two parts making the MS too long, we make a kind request to accept our separate publications.

In tables 5-8, does “non-infection” refer to “no placental infection”?

Response 7: Yes, the word “non-infection” is changed to “no placental infection”.

Tables 5 and 6 can be combined, as can tables 7 and 8.

Response 8: In revised MS, Table 5-8 of the original submitted manuscript are combined to be new Table 5 in the revised manuscript, as suggested by Reviewer 2 and Reviewer 3.

The identification of point mutations is poorly described.  Is one maternal study sequence compared to one reference sequence or to many references or to the corresponding infant sequence?

Response 9: In revised MS, this is specifically described in the paragraph just before Table 6. Please see page 10 lines 307-308.

Table 9 shows nucleotide variations.  How many of these mutations lead to amino acid changes?  Synonymous versus non-synonymous changes should also be discussed in the discussion when point mutations are listed.

Response 10: In revised MS provides more details of the variations, as highlighted at the footnote of the Table 6 and paragraph after table 6. Please see page 14 lines 317-320.

Are the data in Table 9 matched by genotype?  It would be inappropriate to compared across genotypes as this may identify genotype-specific mutations rather than mutations associated with intrauterine transmission.

Response 11: All point mutations observed in this study were not genotype specific. In revised MS we noted this on page 10 lines 312-313.

The statements that “mutation is unlikely to occur during intrauterine infection” and’ . . . no mutation occurred during intrauterine infection process” in the Discussion contradict the rest of the paper . . . particularly Table 9 in which multiple potential mutations are reported.

Response 12: The sentence has been modified by adding the following phrase, “DNA in the pairs of the maternal – cord blood was identical.” to emphasize that no mutation during transmission, as highlighted at the end of the first paragraph of “Discussion” section. Please see page 14 lines 341.

In the discussion section, the authors must reference other studies that have evaluated HBV DNA presence/absence in the placenta.  Are the findings reported here supported by other publications?

Response 13: To the best of our knowledge, no study reported the presence/absence of HBV DNA in the placenta as performed in this study.

The data presented in supplementary figure 3 are not described adequately in the text.  How similar are HBV sequences from the same genotype?

Response 14: In fact, all supplementary figures are mainly the examples of laboratory findings and basic laboratory procedures which the readers are well understand and the omission of these documents does not deteriorate a scientific merit of this manuscript. So, after reconsideration, we decide to delete the supplementary documentations. However, if the reviewers or the editor suggest keeping the documents accompanying with the manuscript we are willing to comply.

Is there sequence data available from the corresponding infants?  If not, how was transmission to the infant documented / confirmed?

Response 15: In revised MS, the data are not available right now. We discussed this point (issue) as one of the limitations of the study and suggest further study, at the end of “Discussion” section. Please see page 17 lines 475-476.

Supplementary figure 6 includes only 6 study sequences.  Where are the remaining ones?

Response 16: The same as “Response 14”

Comments on the Quality of English Language

There are several grammatical errors and awkward phrases throughout the manuscript.  The revised manuscript should be reviewed carefully by a native English speaker and/or a professional editing service.

Response 17: The revised MS has been checked and edited by the native English proofreader, as seen in the attached certificate.

Reviewer 3 Report

            The authors intend to analyze if there is an association between genetic diversity (genotype or specific mutations) of HBV and intrauterine transmission. They state in the Abstract that there is a significant association between genotype B and a higher risk of intrauterine infection. Although the topic is important, many limitations hamper the acceptance of this manuscript.

1.   The number of samples is relatively low. In addition, only two HBV genotypes were found in the studied group. The authors should make an effort to increase the number of samples.

2.   The authors state in the Abstract that there is a significant association between genotype B and a higher risk of intrauterine infection: however, they showed in Results that the difference did not reach statistical significance, which makes a very unfortunate mistake in the Abstract of this manuscript.

3.   It is known that viral load and HBeAg presence are associated to vertical transmission. These two parameters were not analyzed and should be analyzed in a larger study.

4.   No information is provided on the outcome of the children: how many became infected, which ones?

5.   Why some real time positive samples were nested PCR negative?

6.   There are too many tables that could be synthetized in few ones (omitting the negative results (Tables 5 to 8). Table 2 could be transformed in a more compact Figure.

7.   The authors did not demonstrate that the placenta could be a replication site. Their finding only suggests this.

8.   The New Insights on genotypes are only speculative and wrong.

9.   The conclusions again state wrong findings.

Ok.

Author Response

Reviewer Comments

Reviewer 3 (highlighted in purple)

Comments and Suggestions for Authors

The authors intend to analyze if there is an association between genetic diversity (genotype or specific mutations) of HBV and intrauterine transmission. They state in the Abstract that there is a significant association between genotype B and a higher risk of intrauterine infection. Although the topic is important, many limitations hamper the acceptance of this manuscript.

  1. The number of samples is relatively low. In addition, only two HBV genotypes were found in the studied group. The authors should make an effort to increase the number of samples.

Response: Because of several limitations, we could not extend the study at this time. So we have discussed about a relatively small sample size as a weakness of this in “Discussion” section, please see page 17 lines 472-474. Regarding only two genotypes found in this study, in fact, genotype B and C are the two genotypes mainly circulating in the general population of Thailand (1-8).

  1. The authors state in the Abstract that there is a significant association between genotype B and a higher risk of intrauterine infection: however, they showed in Results that the difference did not reach statistical significance, which makes a very unfortunate mistake in the Abstract of this manuscript.

Response: In fact, the relative risk of genotype B had a higher relative risk of 1.4 with 95% CI that does not include 1 or not cross 1. Therefore, it can be concluded that this is significant, though p-value might not less than 0.05. However, because of contradictory values of significance based on RR and p-value, in revised MS, we tone down the conclusion in “Abstract” to have a trend of an increased risk of intrauterine infection, as highlighted. Please see page 1 lines 34.

  1. It is known that viral load and HBeAg presence are associated to vertical transmission. These two parameters were not analyzed and should be analyzed in a larger study.

Response: We agree with the reviewer’s comment that the viral load / HBeAg are associated with vertical transmission. However, this study we focus on HBV genotype and intrauterine infection, as a primary objective, which has never been studied. Additionally, a relatively small number of genotype B, multivariate analysis of HBV genotype with viral load / HBeAg positivity is unlikely to yield  a reliable results.

However, we have discussed this point in the “Discussion” section, we add suggested studies item (1) to determine whether combination of genotype, viral load or positivity of HBeAg can improve the accuracy in predicting vertical transmission. Please see page 17 lines 477-479.

  1. No information is provided on the outcome of the children: how many became infected, which ones?

Response: We agree with reviewer that this is an important information. It is unfortunate that we did not follow-up the mothers and children who enrolled in this study. However, we have discussed this point in the “Discussion” for future study. Please see page 17 lines 479-480.

  1. Why some real time positive samples were nested PCR negative?

Response: Nested PCR is less sensitive than real time PCR, it can detect when viral load of greater than 3 log copies/ml. However, nested PCR is helpful in subgenotype identification.

  1. There are too many tables that could be synthetized in few ones (omitting the negative results (Tables 5 to 8). Table 2 could be transformed in a more compact Figure.

Response: In revised MS, Table 5-8 of the original submitted manuscript are combined to be new Table 5 in the revised manuscript, as suggested by Reviewer 2 and Reviewer 3.

However, we make a request to keep Table 2 as the way it is, since it is easier for readers to see the correlation between positivity tests in mothers, placentas and fetuses.

  1. The authors did not demonstrate that the placenta could be a replication site. Their finding only suggests this.

Response: The placenta could possibly be a replication site, based on the finding of placental infection while no DNA in maternal blood could be detected in some cases, it is therefore only suggestive evidence.

  1. The New Insights on genotypes are only speculative and wrong.

Response: As response earlier (item 2), we tone down the statement from “significance” to “trend”. The conclusion is based on the evidence of relative risk, though not conclusive but this provide the guide or direction for future study to confirm.

  1. The conclusions again state wrong findings.

Response: As response earlier (item 2), we tone down the statement from “significance” to “have a trend of a higher risk of intrauterine infection”, as highlighted in “Conclusion”. Please see page 17 line 486.

Comments on the Quality of English Language : Ok.

Reference

  1. Tangkijvanich P, Mahachai V, Komolmit P, Fongsarun J, Theamboonlers A, Poovorawan Y. Hepatitis B virus genotypes and hepatocellular carcinoma in Thailand. World J Gastroenterol. 2005;11(15):2238-43.
  2. Barusrux S, Nanok C, Puthisawas W, Pairojkul C, Poovorawan Y. Viral hepatitis B, C infection and genotype distribution among cholangiocarcinoma patients in northeast Thailand. Asian Pac J Cancer Prev. 2012;13 Suppl:83-7.
  3. Sugauchi F, Chutaputti A, Orito E, Kato H, Suzuki S, Ueda R, et al. Hepatitis B virus genotypes and clinical manifestation among hepatitis B carriers in Thailand. J Gastroenterol Hepatol. 2002;17(6):671-6.
  4. Chamni N, Louisirirotchanakul S, Oota S, Sakuldamrongpanish T, Saldanha J, Chongkolwatana V, et al. Genetic characterization and genotyping of hepatitis B virus (HBV) isolates from donors with an occult HBV infection. Vox Sang. 2014;107(4):324-32.
  5. Yimnoi P, Posuwan N, Wanlapakorn N, Tangkijvanich P, Theamboonlers A, Vongpunsawad S, et al. A molecular epidemiological study of the hepatitis B virus in Thailand after 22 years of universal immunization. J Med Virol. 2016;88(4):664-73.
  6. Suwannakarn K, Tangkijvanich P, Thawornsuk N, Theamboonlers A, Tharmaphornpilas P, Yoocharoen P, et al. Molecular epidemiological study of hepatitis B virus in Thailand based on the analysis of pre-S and S genes. Hepatol Res. 2008;38(3):244-51.
  7. Theamboonlers A, Tangkijvanich P, Pramoolsinsap C, Poovorawan Y. Genotypes and subtypes of hepatitis B virus in Thailand. Southeast Asian J Trop Med Public Health. 1998;29(4):786-91.
  8. Jutavijittum P, Yousukh A, Jiviriyawat Y, Kunachiwa W, Toriyama K. Genotypes of hepatitis B virus among children in Chiang Mai, Thailand. Southeast Asian J Trop Med Public Health. 2008;39(3):394-7.

Round 2

Reviewer 1 Report

Improvement seen. There is, however, one thing that needs improvement. There  should be more descriptions on the legends of the tables. Scientists and health professionals often have no time to read papers thoroughly. Instead they rely on figures and tables for quick reading. That is why greater descriptions in the legends are imporatant.

Author Response

Comments and Suggestions for Authors

Improvement seen. There is, however, one thing that needs improvement. There  should be more descriptions on the legends of the tables. Scientists and health professionals often have no time to read papers thoroughly. Instead they rely on figures and tables for quick reading. That is why greater descriptions in the legends are important.

Response: Table legends have been slightly modified. However, to avoid repetitive description, we make a kind request to keep table head in succinct form as a standard journal table since short description of the tables are also already presented in text just before the tables. However, if the reviewer or the editor strongly recommend to extend the legends, we are willing to comply.

Reviewer 2 Report

The authors have not provided a response to the previous comments.

Many of the previous comments have been addressed.  However, the authors are not showing a phylogenetic tree.  This is essential for demonstrating that maternal and fetal HBV sequences are identical as stated by the authors.

Moreover, the statement on lines 320-322 about 52 point mutations needs more information.  Presumably, the authors mean 52 point mutations across maternal sequences (since the infant samples are identical to the mother sequences and therefore have no point mutations that distinguish them).  Additional, the ORF in which these point mutations occur should be stated.  Do they occur more than once?

The authors have not explicitly stated that the infant sequences are available. GenBank accession numbers for maternal and infant sequences should be provided.

Line 319 should state "There are four common silent mutations . . . "

Several long, awkward sentences remain and should be revised for readability.

Author Response

Comments and Suggestions for Authors

The authors have not provided a response to the previous comments.

Many of the previous comments have been addressed. However, the authors are not showing a phylogenetic tree.  This is essential for demonstrating that maternal and fetal HBV sequences are identical as stated by the authors.

Response: The phylogenetic tree was constructed and added to the revised-manuscript. Please see page 15-16.

Moreover, the statement on lines 320-322 about 52 point mutations needs more information.  Presumably, the authors mean 52 point mutations across maternal sequences (since the infant samples are identical to the mother sequences and therefore have no point mutations that distinguish them).  Additional, the ORF in which these point mutations occur should be stated.  Do they occur more than once?

Response: We added the information that the 52 point mutations were found in maternal blood, along with the ORF in which these point mutations were occurred. Please see page 14 lines 326-331.

The authors have not explicitly stated that the infant sequences are available. GenBank accession numbers for maternal and infant sequences should be provided.

Response: The GenBank accession number are MK616477-MK616519 and OR061465-OR061470. We added this information on page 4 lines 184-185.

Line 319 should state "There are four common silent mutations . . . "

Response: We changed the word as suggested. (page 14 lines 325 of the revised MS)

Comments on the Quality of English Language

Several long, awkward sentences remain and should be revised for readability.

Response: We understand that although the English usage in this manuscript had been edited by English native speaker who works for our Medical School as indicated by the certificate of editing attached herewith, several long awkward sentences remain to be revised. We have read the revised manuscript again and tried to rephrase the awkward sentences throughout the manuscript in order to make it clear and concise as much as we can. If there are any sentences remains to be rephrased, we would appreciate it very much if the reviewer could help us on this regard to improve this manuscript.

Reviewer 3 Report

The aauthors addressed the concerns.

Author Response

Comments and Suggestions for Authors

The authors addressed the concerns.

Response: Thank you very much.